# Limited and localized magmatism in the Central Atlantic Magmatic Province

R. E. Marzen [1✉], D. J. Shillington [1,6], D. Lizarralde[2], J. H. Knapp[3], D. M. Heffner[4,7], J. K. Davis[1,8] & S. H. Harder[5]

The Central Atlantic Magmatic Province (CAMP) is the most aerially extensive magmatic event in Earth's history, but many questions remain about its origin, volume, and distribution. Despite many observations of CAMP magmatism near Earth's surface, few constraints exist on CAMP intrusions at depth. Here we present detailed constraints on crustal and upper mantle structure from wide-angle seismic data across the Triassic South Georgia Rift that formed shortly before CAMP. Lower crustal magmatism is concentrated where synrift sedimentary fill is thickest and the crust is thinnest, suggesting that lithospheric thinning influenced the locus and volume of magmatism. The limited distribution of lower crustal intrusions implies modest total CAMP volumes of 85,000 to 169,000 km$^3$ beneath the South Georgia Rift, consistent with moderately elevated mantle potential temperatures (<1500 °C). These results suggest that CAMP magmatism in the South Georgia Rift is caused by syn-rift decompression melting of a warm, enriched mantle.

[1] Lamont-Doherty Earth Observatory of Columbia University, Palisades, NY 10964, USA. [2] Woods Hole Oceanographic Institution, Woods Hole, MA 02543, USA. [3] Oklahoma State University, Stillwater, OK 74078, USA. [4] University of South Carolina, Columbia, SC 29208, USA. [5] University of Texas El Paso, El Paso, TX 79902, USA. [6] Present address: Northern Arizona University, Flagstaff, AZ 86011, USA. [7] Present address: Occidental Petroleum Corporation, Houston, TX 77046, USA. [8] Present address: Drummond Carpenter, PLLC, Orlando, FL 32801, USA. ✉email: rmarzen@ldeo.columbia.edu

The Central Atlantic Magmatic Province (CAMP) is the most aerially extensive but one of the most poorly understood large igneous provinces (LIPs) in Earth's history. CAMP magmas have been observed on four continents, extending along eastern North and South America and western Europe and Africa[1]. High-precision radiometric dates suggest this widespread event occurred in multiple episodes over only 600,000 years[2]. Because CAMP magmatism occurred shortly before the End-Triassic extinction[3], the associated degassing and the resulting climate change[4] have been linked to one of Earth's most significant mass extinctions[5,6]. Furthermore, CAMP magmatism has been implicated as an important driver of continental rifting and the breakup of Pangea[1].

The cause of CAMP magmatism remains the subject of significant debate. Although early work hypothesized CAMP might have been caused by a mantle plume[7–9], the absence of a plume trail[10], relatively cool mantle temperatures estimated for CAMP[11], and isotopic and trace element characteristics[12] argue against a plume source and distinguish CAMP from other LIPs. Instead, CAMP may have been caused by delaminated lithosphere followed by mantle upwelling[12,13], edge-driven convection[10,14,15], and/or elevated mantle temperatures from tens of millions of years of continental insulation[16,17].

Despite the significance of this magmatic event, the volume and distribution of CAMP magmas throughout the crust, and thus the total magnitude of the event, are poorly known. CAMP is estimated to have a volume of ~3 million km$^3$ based on analysis of shallow intrusions and lavas onshore and assumptions on fractionation[18]. CAMP is thought to be a relatively low-volume LIP because the average thickness of magmatic addition of ~0.3 km[18] is far less than estimates for other major LIPs, which are typically ~1 km or greater[19,20]. There are, however, very few direct or indirect observations on the volume and distribution of CAMP intrusions in the Earth's mid- to lower crust[21,22], which are needed to constrain these estimates and evaluate competing models for its cause. Another uncertainty in estimating the total volume of CAMP is the age and origin of magmatism offshore along the rifted margins of Pangea. Although extensive magmatism has been imaged on these margins including in the Blake Plateau Basin and the Carolina Trough[23,24], the timing and duration of the emplacement of this magmatism are unknown[25]. Ages between 172 and 200 Ma[26–30] and emplacement durations up to 6–31 Myr[31] have been suggested, so it is unclear if offshore magmatism is related to CAMP.

The southeastern United States (SE US) is an ideal location to characterize the subsurface volume and distribution of CAMP magmatism and controls on its emplacement. This region lies within the known extent of CAMP, and CAMP dikes have been dated and characterized within the southeastern US[12,32]. The South Georgia Rift Basin formed ~235–205 Ma[33–36] and is the largest of the Triassic rift basins along the Eastern US. Formation of the South Georgia Basin was followed by the emplacement of CAMP magmatism ~201 Ma[32], and ultimately the breakup of Pangea ~175–195 Ma[26–28]. Tectonic sutures that formed during multiple stages of Appalachian orogenesis before rifting[37,38] helped localize extension and the formation of rift basins across eastern North America[33], including the South Georgia Rift Basin[39].

In the following sections, velocity models on two seismic transects that cross the South Georgia Rift (Fig. 1) are used to constrain CAMP magmatism at depth and evaluate the relationship between magmatism and Triassic extension. Our results indicate that there are modest volumes of mafic magmatic intrusions, which concentrate in the western portion of the South Georgia Rift, where the thickest synrift sedimentary fill and the most crustal thinning is observed. The locus and distribution of

these intrusions in the South Georgia Rift are consistent with decompression melting at the somewhat elevated mantle potential temperatures associated with CAMP[11,12]. These findings suggest that synrift decompression melting may explain the volume and distribution of lower crustal magmatic intrusions in the South Georgia Rift.

## Results

**Velocity model constraints on crustal structure.** P-wave velocity models based on wide-angle seismic reflection/refraction data acquired along two profiles across the South Georgia Basin during the SUwanee Suture and GA Rift basin experiment (SUGAR) constrain the depth of basin fill, crustal thickness, and the volume and distribution of CAMP magmatic additions (Fig. 2). We identified refractions through the sedimentary fill, crust, and upper mantle, and reflections off the base of the sedimentary basin (Line 1 only) and the Moho. Travel-time picks of these phases were used to invert for P-wave velocity structure of the sedimentary basins, crust, and upper mantle using the code VMTomo[39–41] (Methods).

Both seismic profiles indicate limited and localized regions of elevated >7.0 km s$^{-1}$ lower crustal velocities in the South Georgia Rift (Fig. 2a, b). The most likely explanation for changes in lower crustal velocity in this region is a change in composition. The observed variations are within a single crustal terrane[38], so contrasts between crustal terranes cannot explain our observations (Fig. 1). We thus interpret these localized increases in lower crustal velocity as the addition of mafic magmatic intrusions[42,43]. Seismic refraction measurements from offshore of eastern North America indicate that mafic lower crustal velocities typically range from 7.2 to 7.5 km s$^{-1}$[24,44–46], which is similar to the highest lower crustal velocities directly constrained by rays that turn in the lower crust in the SUGAR velocity models. These velocities also encompass different intrusion compositions predicted for different depths of melting[47]. In contrast to velocities of mafic intrusions, unmodified continental lower crust is typically ~6.8 km s$^{-1}$[48], and well-constrained lower crustal velocities on SUGAR Line 2 indicate that the lower crustal velocities northwest of the South Georgia Rift Basin are ~6.75 km s$^{-1}$[39].

On Line 2 across the eastern South Georgia Rift, the lower crust is almost uniformly <7 km s$^{-1}$, implying limited to absent mafic addition to the lower crust. The crust thins abruptly from 38 to 32 km over a distance of 40 km centered at 150 km on the Line 2 transect, which is likely controlled by the Alleghanian suture serving as either a pre-existing weak zone or rheological boundary between Laurentian and Gondwanan crust[39]. In contrast, on Line 1 across the western South Georgia Rift, >7.0 km s$^{-1}$ lower crustal velocities are observed in the center of the seismic line, but decrease to <7.0 km s$^{-1}$ to either side (Fig. 2a, b). The high lower crustal velocities on Line 1 coincide with the thickest syn-rift sedimentary fill and a shallowing of the Moho (Fig. 2a, b). Rift basin sedimentary fill is thicker on Line 1 than on Line 2, which is consistent with seismic reflection imaging and core data from the western versus eastern South Georgia Rift (Fig. 1)[25,49] Thus, we observe a correlation between the thickness of interpreted mafic lower crustal intrusions and the amount of crustal thinning associated with formation of the South Georgia Rift.

**South Georgia Rift magmatism.** The most striking observation from our velocity models is the localization of lower crustal mafic magmatic intrusions, which contrast with the widespread distribution of CAMP at the surface[50–52]. The correlation between magmatic intrusions and the Triassic South Georgia Basin evident in velocity models is surprising because multiple geological constraints suggest that magmatism was emplaced after, not

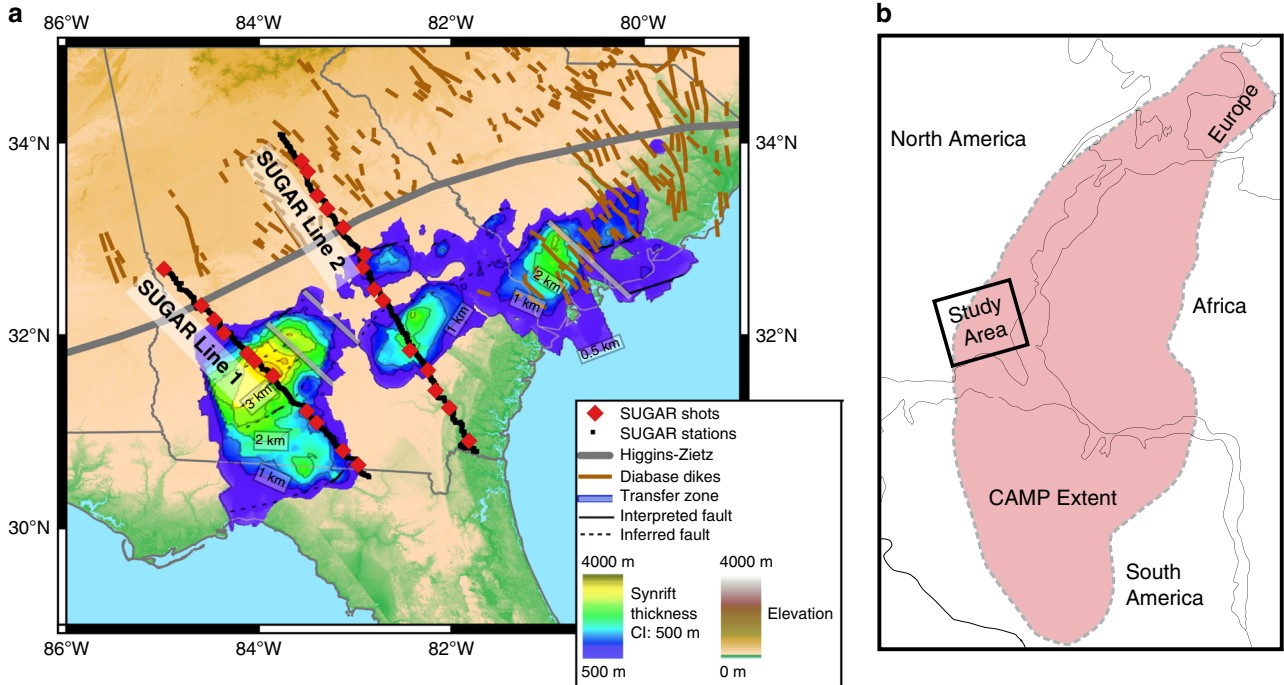

**Fig. 1 Map of the South Georgia Rift Basin in the southeastern United States. a** Locations of Lines 1 and 2 from the SUGAR refraction seismic experiment, the Higgins-Zietz magnetic boundary interpreted as the Alleghanian suture[38,69], CAMP dikes mapped via magnetic data[50] and field observations[51], and the isopach map of synrift sedimentary fill in the South Georgia Basin constrained by well and seismic reflection and refraction data[49]. SUGAR seismic shots are indicated with red diamonds and receivers are shown with the dark line beneath the shots. **b** The approximate extent of CAMP shaded in red over a reconstruction of Pangea[18].

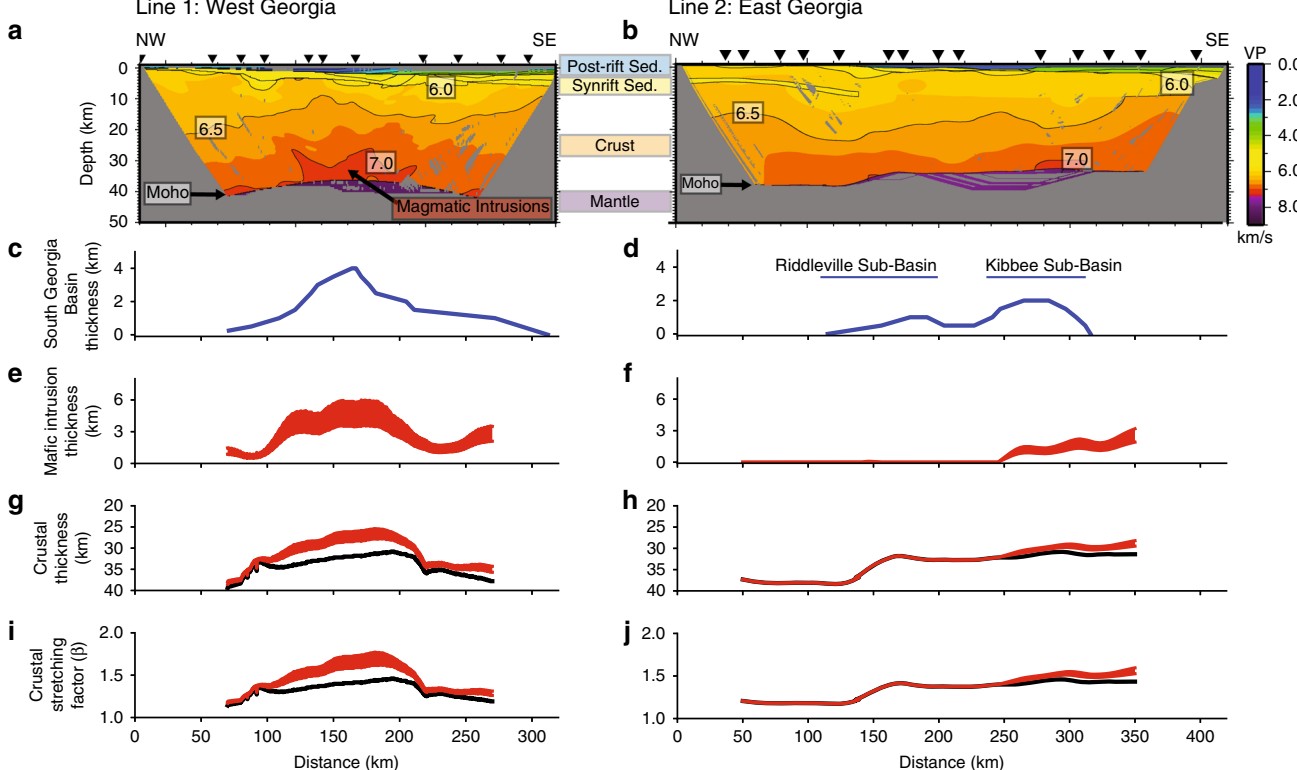

**Fig. 2 Crustal stretching and magmatism inferred from velocity models. a, b** $V_P$ models with 0.5 km s$^{-1}$ contour interval for Line 1 (**a**) and Line 2 (**b**). Black triangles indicate shot locations. Areas with no ray coverage are masked in gray. **c, d** South Georgia Basin thickness with labels for sub-basins beneath Line 2[49]. **e, f** Range of possible thicknesses of lower crustal mafic intrusions assuming mafic intrusion velocities from 7.2 km s$^{-1}$ to 7.5 km s$^{-1}$. **g, h** Observed crustal thickness (black) and crustal thickness without estimated lower crustal mafic intrusions (**e, f**). **i, j** Crustal stretching estimates assuming an initial crustal thickness of 45 km (black)[55]; red curves show crustal thickness and crustal stretching factor with mafic intrusions (**e, f**) removed.

during, the formation of the South Georgia Rift. First, CAMP-dated sills and dikes crosscut synrift strata and orogenic structures in the shallow subsurface[33,53]. Second, flows in the Southeastern US are only found in the postrift section[35,36]. Third, the orientation of CAMP-dated dikes (Fig. 1) is incompatible with the NW-SE minimum horizontal stress orientation consistent with the orientation of basin structures[34]. The new observations presented here suggest a relationship between lithospheric thinning during the formation of the South Georgia Rift and magmatism that was not previously recognized, but this relationship raises a number of questions. Magmatism connected to rifting would be expected to have occurred contemporaneously with rifting at ~230–205 Ma. In contrast, all available evidence from the stratigraphic record[33,53], dike orientations as an indicator of the stress field[34], and intrusion dates in the Southeastern US[32,54] indicate that near-surface magmatism in the Southeastern US occurred during CAMP ~201 Ma, after South Georgia rifting. We evaluate the conditions that might explain this limited and localized magmatism and its possible connection to extension to form the South Georgia Basin by estimating the volume of magmatism from the seismic velocity models in comparison to the volume of intrusions expected during synrift decompression melting.

To quantify the amount of magmatism in this region, we used a linear mixing calculation to parcel observed lower crustal velocities into an unmodified, lower velocity component and a higher velocity component characteristic of mafic magmatic intrusions. Because the composition and velocity of magmatic intrusions here are unknown, we calculated intrusion thicknesses using end-member possible velocities of 7.2 and 7.5 km s$^{-1}$ (Methods)[43]. On SUGAR Line 1, the elevated lower crustal velocities at the center of the line are consistent with ~3–6 km of mafic intrusions in the lower crust (Fig. 2e). In contrast, the largely < 7.0 km s$^{-1}$ lower crustal velocities along Line 2 translate to ~0–3 km of mafic intrusions, with the largest inferred intrusion volumes located towards the SE end of the line at the Georgia coast, approaching the Atlantic rifted margin (Fig. 2f).

The patterns of crustal thinning, magmatic addition, and extent of the South Georgia Rift Basin on Line 2 suggest that two episodes of rifting concentrated crustal thinning along different portions of the transect. We infer that thinning of the crust beneath the South Georgia Basin (Fig. 1) was associated with the extension to form this rift basin and that the southeastern portion of Line 2 was likely additionally thinned later during the breakup of Pangea ~175–195 Ma[26–28,39]. Although the exact timing of continental breakup is uncertain, we would expect this successful rifting event to concentrate crustal stretching and magmatic intrusions towards the margin. This is consistent with the pattern of crustal stretching and magmatic addition from 280 km distance to the southeastern end of the transect, which increases towards the rifted margin and is not centered beneath the South Georgia Rift. To focus our analysis on the South Georgia Rift, the analysis below excludes the southeastern portion of Line 2.

The spatial correlation between areas with elevated lower crustal velocities and the extent of the South Georgia Rift Basin suggests that extension influenced the generation and emplacement of magmatism. To evaluate the spatial relationship between crustal thinning and magmatism, we use our velocity models to quantify the amount of crustal stretching, which we express as the β factor (initial thickness)/(extended thickness). Extended crustal thickness was measured from the base of the sedimentary fill to the Moho (Fig. 2). We assume a prerift crustal thickness of 45 km, consistent with average continental crust and modern crustal thicknesses at the current boundary between the Appalachians and the Coastal Plain[48,55]. Because rifting occurred within a relatively young orogen, crustal thicknesses were likely greater than 45 km, in which case modern crustal thicknesses outside the

rift basin provide a lower estimate of β than for an initially thicker crust. The amount of crustal stretching, without considering magmatic addition, is up to ~1.4 (black line, Fig. 2i, j). Higher stretching factors are obtained if estimated magmatic intrusions are removed (red line, Fig. 2i, j). On Line 2, where magmatic intrusions into the lower crust appear to be limited, β gradually increases from ~1.2 in the NW to ~1.4 in the SE. On Line 1, the area of greatest crustal thinning (β ~1.4–1.8) occurs beneath the South Georgia Basin and coincides with significant magmatic intrusion thickness and a thicker synrift sedimentary fill[49] (Fig. 1).

We used the positive correlation between synrift sediment thickness and lower crustal mafic magmatic intrusion thickness observed on SUGAR refraction profiles to estimate the total volume of lower crustal magmatic intrusions beneath the South Georgia Basin (Methods). Synrift sediment thickness across the South Georgia Basin is based on seismic reflection data, seismic refraction data, and well data[49] (Supplementary Note 1). From this correlation, we infer between 76,000 and 127,000 km$^3$ of lower crustal mafic magmatic intrusions assuming a lower crustal intrusion $V_P$ of 7.5 and 7.2 km s$^{-1}$, respectively. We then estimate the volume of upper crustal CAMP magmatism based on the extent of the South Georgia Basin and the typical range of thicknesses of basalt and diabase layers in well data of ~100 to 500 m[25,56]. Near-surface CAMP volumes are between 8000 and 42,000 km$^3$ given the basin area of ~83,000 km$^2$ (Fig. 1). This estimate does not include upper crustal intrusions beneath the basins that may be elevating upper crustal seismic velocities (i.e., 100–150 km distance on Line 1, Fig. 2) or dikes in the region for which the depth extent is unknown[50,51]. Combined, this gives an estimated total volume of magmatic addition associated with South Georgia Rift of ~85,000 to 169,000 km$^3$. This implies an average melt thickness of ~1.5 km in the South Georgia Rift Basin, where magmatism appears to be particularly concentrated compared to areas within the aerial extent of CAMP but outside of the rift basin.

**Conditions during magma generation and emplacement.** The observed spatial relationship between magmatic intrusions and crustal thinning is consistent with decompression melting, where melt production is promoted by lithospheric thinning. This observation motivates us to compare seismic constraints on the volume and distribution of magmatism with predicted magmatism from decompression melting at different mantle potential temperatures. These models provide a means to compare the quantity of lower crustal magmatic intrusions to volumes predicted to be generated during rift-related decompression melting because we cannot constrain the timing or source of lower crustal magmatic intrusions directly. We used a batch melting model[57] to calculate the melt fraction given a specified pressure, temperature, and mantle composition (Methods). For mantle potential temperatures ranging from 1300 to 1500 °C, the vertical melt fraction resulting from decompression due to lithospheric thinning was integrated to determine melt thickness.

For a baseline model with a typical mantle potential temperature of ~1350 °C and uniform lithospheric stretching, we would expect less than 1 km of intrusions across both SUGAR seismic lines, which is significantly lower than the interpreted thicknesses on SUGAR Line 1 (Fig. 2). The magmatism and stretching factors along the SUGAR seismic transects are consistent with decompression melting with modestly elevated mantle potential temperatures of 1425–1475 °C, which is similar to geochemically based estimates of mantle potential temperature for CAMP[11,12]. These calculations largely preclude very high mantle temperatures (e.g., >1500 °C), which would produce larger

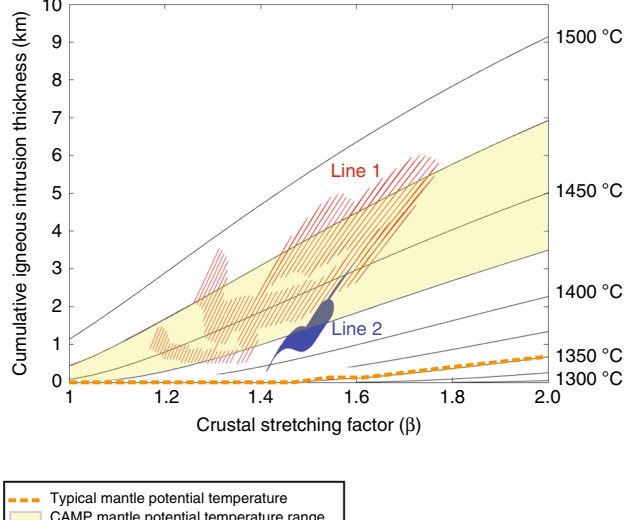

**Fig. 3 Decompression melting models compared to seismic constraints.**
Predicted magmatic intrusion thickness assuming uniform lithospheric stretching at a range of mantle potential temperatures (black lines and text). Red and blue lines show the estimated range of magmatic intrusion thicknesses from SUGAR velocity models (Fig. 2). The orange line shows expected intrusion thickness for a baseline scenario with a normal mantle potential temperature of 1350 °C. The yellow shaded area shows range of mantle potential temperatures for CAMP from geochemical constraints[11,12].

volumes of magma than indicated by the observed lower crustal velocity structure (Fig. 3). These calculations demonstrate that decompression melting with modestly elevated mantle potential temperatures could explain the observed distribution of magmatism and extension in the South Georgia Rift. Insulation below the Pangea supercontinent is expected to have elevated temperatures as much as 100 °C[16], enough to produce the observed magmatism by this mechanism. Synrift magmatism would have also been promoted by other factors that were not included in the decompression melting calculations, particularly the enrichment of the mantle below Pangea by prior subduction[13,58].

## Discussion

The combined geophysical, geological, and geochemical constraints on magmatism in the Southeastern US can be explained by decompression melting and emplacement of lower crustal intrusions during continental extension between 230 and 205 Ma. The geochemistry of CAMP magmas indicates 30–50% fractionation from a mantle-derived magma[18], which is consistent with our new constraints on the amount of magmatism at depth. However, the biggest challenge in reconciling available constraints is the timing of emplacement of magmatism. The amount of magmatism emplaced in the lower crust has a similar volume and spatial distribution to what would be expected for decompression melting during rifting (~230–205 Ma) of a warm mantle, but rifting in the Southeastern US took place earlier and over a longer time period than the short ~0.6 million year duration of CAMP intrusions at 201 Ma. Explaining this timing difference requires that some mechanism, such as a change in stress state[59] (i.e., associated with extension at the modern rifted margin), allowed magmatism generated by synrift decompression melting below the South Georgia rift to be emplaced at the Earth's surface over a short period of time. In some active rift systems, evidence for magmatism is observed at depth even though volcanism is not

observed at the surface[60]. In the South Georgia Rift, the relatively thick crust remaining after limited continental extension may have delayed transport of magmatism to the Earth's surface.

Our data do not rule out the possibility that decompression melting of a warm, enriched mantle produced magmatism beneath the South Georgia Rift, and CAMP was caused by a later "event", such as delamination[13], although this interpretation is difficult to reconcile with estimates of the fractionation of CAMP magmas. If such an event produced significant volumes of magmatism beneath the South Georgia Rift, intrusions must have been localized by pre-thinned lithosphere[61,62].

These new constraints indicate that volumes of CAMP magmatism were limited. In a high-end scenario case, if one assumed that all of the lower crustal mafic magmatic intrusions in the South Georgia Rift are associated with CAMP, CAMP volumes in the South Georgia Rift constitute only 2–6% of the total 3-million-km³ volume estimated for CAMP as a whole[18]. More constraints on crustal intrusions are needed to refine estimates of the total volume of CAMP, but our results contribute to the growing body of evidence that CAMP is different from other known LIPs. Previous studies have shown that mantle potential temperatures were relatively cool for a LIP[11] and that the contribution of a deep mantle component to magmas was limited[12]. Our work adds to this picture by indicating a limited total volume for CAMP throughout the crust and thus rules out mechanisms for the generation of CAMP that result in large total volumes of magmatism. Nonetheless, CAMP appears to have resulted in a major biotic crisis at the end of the Triassic. New constraints from this study on the volume and distribution of magmatic intrusions at depth can be used to evaluate the contribution of intrusive magmatism from CAMP to $CO_2$ degassing and environmental change[3,4,63].

## Methods

**Seismic data and tomographic inversion.** SUGAR Line 1 in the western South Georgia Rift was collected in March 2014 and included 11 explosion sources recorded by 1193 geophones spaced ~250 m apart. SUGAR Line 2 in the eastern South Georgia Rift was collected in August 2015 and included 14 explosion sources recorded by 1981 geophones ~200 m apart (Fig. 1; Supplementary Fig. 1; Supplementary Data 1).

To enable comparison between the two profiles, consistent seismic data processing, phase identification and velocity modeling were applied to SUGAR Lines 1 and 2. A detailed description of the Line 2 shots and data and analysis of SUGAR Line 2 is described by Marzen et al.[39]. Supplementary Table 1 lists information about the Line 1 shot locations, timing, and charge sizes. The processing steps were bandpass filtering the shot gathers at 3–14 Hz, applying offset-dependent gains and amplitude normalization to traces, and applying a reduction velocity of 8 km s⁻¹ to facilitate identification of seismic arrivals[64]. In the processed shot gathers, we observe clear arrivals at offsets up to 320 km (the total length of SUGAR Line 1). Interpreted phases included refractions through the sedimentary fill, crust, and upper mantle, and reflections off the base of the sedimentary basin (Line 1 only) and the Moho. Pick errors were assigned based on confidence in the arrival interpretation, and generally varied between 0.04 and 0.15 s, though larger uncertainties were assigned to small subsets of picks (e.g., at very far source-receiver offsets or in areas of complex shallow structure). Interpreted arrivals, shot, and instrument data are provided in Supplementary Data 1.

In records from shots located within the South Georgia Basin (Shots 4–14, Supplementary Figs. 1, 2), two clear sedimentary refractions are observed with distinct apparent velocities. At offsets less than ~5 km, sedimentary refractions have an apparent velocity of ~2–2.5 km s⁻¹. At offsets between ~5–20 km, sedimentary refractions have apparent velocities of ~4.5–5 km s⁻¹. Reflections were identified between these sedimentary layers and from the base of the sediments (e.g., Supplementary Fig. 2). For shots north of the South Georgia Basin (Shots 1 and 3, Supplementary Fig. 1), sedimentary refractions are absent. Crustal refractions (Pg) are identified as first and secondary arrivals out to offsets up to 250 km; apparent velocities increase with depth from ~6 to >7 km s⁻¹. We observe mantle refractions (Pn) on multiple shots, which exhibit high apparent velocities of >8 km s⁻¹ (e.g., Supplementary Fig. 2). The crossover distance of Pg and Pn is ~180–200 km. PmP arrivals were typically identified at offsets between 80 and 180 km. We picked P-wave arrivals for each of these phases and assigned travel-time uncertainties by visual inspection (Supplementary Table 2). Supplementary Fig. 3 shows additional images of interpreted phases on Line 1, and similar images for Line 2 are in Figs. S1–S14 from the Supporting Information of Marzen et al.[39]. A

comparison of shot gathers from the two profiles illustrates the differences in velocity structure (Supplementary Fig. 2). On the shot gather from SUGAR Line 1, sedimentary refractions are observed to larger source-receiver offsets, reflecting the thicker synrift sediment in this part of the South Georgia Basin. Additionally, the apparent velocities of crustal refractions (Pg) on SUGAR Line 1 are higher than those in SUGAR Line 2, particularly for arrivals at large source-receiver offsets that sample the lower crust.

We modeled travel-time picks of reflections and refractions from the sediments, crust and upper mantle to constrain the P-wave velocity structure. The shots on SUGAR Line 1 were projected onto a two-dimensional line with end points at 30.509°N, 82.833°W and 32.711°N, 85.0104°W, and the shots for Line 2 were projected on a line with end points of 30.743°N, 81.706°W and 34.101°N, 83.760° W. The source-receiver offsets for both lines were taken from the real geometry and assumed to fall along these 2D lines. The sediment basin structure was determined by iterative forward modeling and inversion in RAYINVR using sedimentary reflections and refractions, a well log near Line 2[56], and topography on Pg and Pn caused by shallow structures[65]. This code employs a coarse velocity model parameterization with user defined nodes, which enabled us to incorporate direct constraints on basin structure from sedimentary refractions, indirect constraints from topography on Pg arrivals, and constraints from other datasets (e.g., COCORP reflection data). We then left the basin structure determined from RAYINVR fixed and inverted for the crustal and upper mantle structure using VMTOMO. The forward step of VMTOMO involves ray tracing using the graph method, and the inverse step uses a damped least squares method to minimize a cost function with data misfit and smoothing/damping terms. Multiple iterations of forward modeling and inversion were applied, in which misfit was gradually reduced and smoothing/damping constraints were relaxed to allow structure to emerge. Horizontal smoothing was generally 5 times greater than vertical smoothing. Early inversions for seismic velocity only included near-offset arrivals and thus only updated the upper crust; deeper portions of the model were gradually included by progressively incorporating longer-offset phases[40,41,66].

These models fit the data well, with $\chi^2$ of 1.27 and root mean squared (RMS) misfit of 72 ms for Line 1 (Supplementary Table 2) and $\chi^2$ of 0.90 and RMS misfit of 85 ms for Line 2[39]. The ideal $\chi^2$ value is 1, but a larger value was allowed on Line 1 to avoid introducing small-scale velocity artifacts due to 3D geometry and poorly constrained variations in basin structure. Supplementary Tables 2 and 3 show misfit on Line 1 by shot gather and phase, respectively, and data misfit for all picks is illustrated in Supplementary Fig. 4 for Line 1 and Supplementary Fig. 5 for Line 2. The velocity models for Lines 1 and 2 are provided in Supplementary Data 2 and 3.

The deeper portions of the velocity models including the lower crust are the most challenging to resolve, and there are tradeoffs between increasing crustal thickness and increasing lower crustal velocity. In order to evaluate uncertainty in the velocity of the lower crust, we examined model misfit associated with perturbations in lower crustal velocity and Moho depth (Supplementary Note 2). These tradeoff tests show that the velocity of the lower crust can only be perturbed by up to ~0.05 km s$^{-1}$ without increasing the $\chi^2$ fit to the data beyond an acceptable level (Supplementary Fig. 6 for Line 1 and Supplementary Fig. 7 for Line 2). The data, however, do not resolve the precise dimensions and locations of localized lower crustal velocity perturbations on the scale of tens of km. In addition, our velocity models are most sensitive to perturbations in lower crustal velocity and Moho depth in the central portions of each seismic line where reversed ray coverage is most abundant. In summary, these velocity models are sensitive to overall lower crustal velocity but cannot resolve smaller scale variations in lower crustal velocity. The basis of our result is the large-scale differences in lower crustal velocity and crustal thickness between SUGAR Lines 1 and 2, which are well constrained.

**Igneous intrusion thickness calculations from velocities**. We estimated the thickness of intruded magmas by adapting the linear mixing calculation[43] (Fig. 2):

$$Z_{int} = \max\left(Z_{tot} * \left(V_{P-orig} - V_{P-obs}\right) / \left(V_{P-orig} - V_{P-int}\right), 0\right),$$

where $Z_{int}$ is the thickness of mafic intrusions, $Z_{tot}$ is the thickness of the crust below 20 km depth, $V_{P-orig}$ is the reference velocity for the lower crust without intrusions, $V_{P-obs}$ is the observed average lower crustal velocity (below 20 km depth), and $V_{P-int}$ is the assumed velocity of mafic magmatic intrusions. The average observed lower crustal velocity ($V_{P-obs}$) was calculated from 20 km depth to the Moho across each seismic line. This depth range $Z_{tot}$ was selected because increases in crustal velocity at these depths reflect changes in composition rather than the closure of cracks and pore spaces observed in the shallower crust[48]. The reference velocity for unmodified lower crust was estimated at 6.75 km s$^{-1}$,[39]. The velocity of material that intruded the lower crust was estimated at 7.2–7.5 km s$^{-1}$,[24,44–47]. Because negative intrusion thicknesses are generated when the average velocity of the lower crust is less than 6.75 km s$^{-1}$, $Z_{int}$ in these scenarios is set to 0 km.

**Decompression melting models**. We use the Katz parameterization[57] to calculate melt fraction through a 1D column at a range of depth (pressure) and temperature conditions for different degrees of thinning of the crust and mantle lithosphere. In this calculation, we assume a mantle peridotite composition of 15% anhydrous clinopyroxene by weight[67]. This choice reflects the fact that the degree of mantle enrichment varies within the extent of CAMP but is comparatively low in the SE

US compared to farther north[13]. Were the mantle to have a hydrous composition or include other volatiles from prior subduction, a greater amount of melt would be produced while the volatiles are present in the mantle[57,58].

We assume an initial crustal thickness of 45 km[55] and lithospheric thickness of 120 km[68]. These initial thicknesses are taken from seismic observations to the northwest of our study area where there is neither a deep Appalachian root nor evidence for crustal thinning.

The calculated melt fraction versus depth was converted to igneous crustal thickness for a given mantle potential temperature, crustal thinning, and assumed amount of lithospheric thinning. To calculate the pressure at the lithosphere-asthenosphere boundary for different lithosphere extension scenarios, we assumed a continental crust density of 2800 kg m$^{-3}$ and a mantle lithosphere density of 3300 kg m$^{-3}$. For a given amount of crustal and mantle lithospheric thinning, the thickness of igneous intrusions was determined by integrating the resulting melt fraction over depth.

Another important contribution to expected rift magmatism is the degree of depth-dependent stretching. We consider both a uniform stretching case (Fig. 3) and scenarios where the whole lithosphere has experienced 2x and 4x more extension than the crust (Supplementary Fig. 8):

$$(a - 1) = k * (\beta - 1)$$

for whole-lithosphere stretching factor α and crustal stretching factor β, where whole lithosphere extension is a multiple $k$ of crustal extension.

To accommodate uncertainty in post-orogenic but prerift thickness of the crust and lithosphere, decompression melting calculations for likely end-member crustal (40 km, 55 km) and lithospheric (90 km, 150 km) thicknesses are included in Supplementary Fig. 9. More melt is produced by decompression melting when the initial lithosphere is thinner. The inferred mantle potential temperature is greater when just the initial crust is thinner because the crustal stretching factor is smaller for the same inferred amount of igneous crustal thickness. Our observations are consistent mantle potential temperatures less than 1500 °C for initial lithosphere thicknesses up to 150 km.

In summary, despite uncertainties in initial thickness and depth-dependent stretching, modeling results are consistent with decompression melting and moderately elevated mantle potential temperatures.

**South Georgia Rift magma volume calculation**. We developed an estimate of the volume of CAMP magmatism using (1) the thickness of the sedimentary fill in the South Georgia Basin (Supplementary Note 1)[49] and (2) the thickness of lower crustal mafic magmatic intrusions on SUGAR Lines 1 and 2. This approach is based on the first-order observation that the thickness of magmatic intrusions is greater where synrift sediments are thicker. We calculated the average intrusion thickness (Fig. 2e–f) in 500-m bins of South Georgia Basin synrift sediment thickness (e.g., 2000–2500 m) on SUGAR Lines 1 and 2 (Supplementary Fig. 10). All parts of both seismic lines were used to constrain the calibration except where we do not have resolution of the lower crust and Moho or where the magmatism may be sourced from the breakup of Pangea instead of CAMP at the southeastern end of Line 2 (i.e., constraints from Line 1: 50–250 km distance; Line 2: 50–280 km distance).

We assumed no magmatic intrusions where the South Georgia Basin synrift sediments are less than 1000 m thick, which is consistent with observations from lower crustal velocities where the statistical average was near zero (Supplementary Fig. 10). Where the South Georgia Basin synrift sediments are >1000 m thick, we estimate the volume of magmatism in the lower crust by multiplying the area of the South Georgia Basin within each synrift sediment thickness bin by the average magmatic intrusion thickness for that sediment thickness. The equation below represents how we used the basin model to estimate volumes of lower crustal magmatic intrusions by summing across each bin $i$:

$$M = \sum_{i=1}^{n} a_i \times m_i,$$

where $M$ is the total volume of magmatism, $a$ is the surface area of the South Georgia Basin that falls within a 500-m syn-rift sediment thickness bin, and $m$ is the mean magma intrusion thickness calculated for that bin (or 0 for the 0–500 and 500–1000 m bin). These values are provided in the Supplementary Table 4. From this method, we estimate between 76,000 and 127,000 km$^3$ of mafic magmatic intrusions in the lower crust across the South Georgia Rift. We performed this calculation using a range of bin sizes and found that the resulting estimate of volume is not very sensitive to the choice of bin size.

We make a conservative estimate of the volume of magmatism in the upper crust based on the assumption that the thickness of basalt or diabase layers from well data[25] reflects the range of intrusion thickness within the extent of the South Georgia Basin—between 50 and 500 m. We then multiplied the area of the basin[49] by these two end-member intrusion thicknesses to estimate the volume of magmatism in the near-surface. From this method, we estimate near-surface intrusion thicknesses between 8,300 and 42,000 km$^3$ in the South Georgia Rift. This estimate is similar to the methods used in other calculations of near-surface CAMP volumes[1,4,18], but does not account for intrusions that may exist in the shallow crust beneath or outside the South Georgia Basin.

## Data availability

The SUGAR refraction seismic dataset analyzed in the current study is available on request through the IRIS Data Management Center, report number 14-023, http://ds.iris.edu/ds/nodes/dmc/forms/assembled-data/.

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

## Acknowledgements

This project was funded by an NSF GRFP fellowship DGE 16-44869 and a grant from the National Science Foundation's Division of Earth Sciences (NSF-EAR) EarthScope program through the collaborative awards EAR-1144534/−1144829/−1144391. Data collection was made possible with help from IRIS PASSCAL, the University of Texas El Paso Seismic Source Facility, the teams of students who deployed and recovered geophones, and the support of landowners and county and state officials. We thank Alistair Harding for the VMTomo code, Nathan Miller for the PyVM toolbox, and William Wilcock for maintaining the Upicker package to pick arrivals.

## Author contributions

R.E.M. and D.J.S. drafted the main text and modeled seismic velocities. R.E.M. calculated intrusion thicknesses and magmatism volumes. D.J.S., D.L., and S.H.H. conceived of the experiment, acquired funding, and led data acquisition. D.M.H. and J.H.K. modeled the South Georgia Basin thickness. R.E.M. and J.K.D. undertook decompression melting analysis. D.L., J.H.K, D.M.H., J.K.D., and S.H.H. provided feedback on the manuscript.

## Competing interests

The authors declare no competing interests.
