## [Peer Review File · Nature Communications]

Reviewers' comments:

Reviewer #1 (Remarks to the Author):

This is a very interesting, well-written, important manuscript about geophysical data on the crustal structure in the south-eastern USA (Georgia rift). The authors show quite convincingly that the deep crust was intruded by basic magmas in at least parts of the crustal roots of the South Georgia Rift. The thickness of such basic layers is does generally not exceed 2 km and is sometimes much thinner or even absent. The authors interpret these mafic layers as related to the CAMP event and consider that the generally low volumes of such rocks indicate that CAMP magmatism formed from a not very hot mantle. Apparently, there is also a thinning of the deep crustal gabbros from West to East, i.e. towards the central Atlantic margin. These findings are very important, as they constrain for the first time the deep crustal roots of CAMP magmatism (however, intermediate crust intrusions are present in west Africa, i.e. Sierra Leone; Callegaro et al., *J. Petrology*, 2017) and bear important constraints on the genesis of this magmatic province and on its relation with the break-up of Pangea and the opening of the Atlantic ocean. The conclusions are compatible with what we know from the rest of CAMP. Figures are very nice. The adopted methods are excellent, as far as I can understand as a non-specialist (geophysicist). The only part of the manuscript, which seems little convincing to me is the discussion about the timing of this basic magmatism. The authors suggest that this magmatism may be significantly older (by at least 4 million years) than the rest of CAMP. This is based chiefly on the fact that sedimentation in the South Georgia Rift stopped by the end of the Norian (ca. 205 Ma). However, the evidence for such stop of sedimentation and thus rifting is not clear from the text. The authors should briefly discuss geological evidences of this. The authors seem also to propose that the deeply intruded magmas may be significantly older than those erupted or intruded as shallow sills or dykes. Such idea is not very convincing, as in any case a shallow magma requires the presence of deep intrusions. I.e., CAMP basalts are evolved melts resulting from at least 40-50% fractionation of a primary mantle melt. The fractionated mass and volume intruded in the intermediate or deep crust and may well be the parts of the basic intrusions identified by the authors.

All the best,

Andrea Marzoli

Reviewer #2 (Remarks to the Author):

I have reviewed the manuscript by Marzen and colleagues entitled, "Limited and Localized Magmatism at the Heart of the Central Atlantic Magmatic Province."

The authors have used velocity models on two seismic transects from the southeastern United States to constrain the volume and distribution of CAMP-related crustal intrusions. They conclude that the volume of igneous intrusions is small and that the intrusive activity was focused beneath depocenters of the South Georgia rift basin complex. Thus, crustal thinning associated with Triassic rifting affected the magnitude and distribution of CAMP-related intrusive activity.

It is very interesting to see convincing (and quantifiable) evidence of CAMP-related crustal intrusions beneath an onshore rift basin on the margin of eastern North America. It is also very interesting to see that the shallow distribution of CAMP-related igneous activity is much broader than that in the lower crust.

As with most research, this work answers some key questions and raises many more. I think that the research will be of interest to geologists and geophysicists who study rifted margins worldwide,

not just the rifted margin of eastern North America.

As discussed below, I found some parts of the manuscript difficult to read and/or understand. Also, some of the figures need improvement and clarity.

Figure 1.

I have some issues with this figure and caption. The authors need to describe the insert on the bottom right. I presume that this is a Triassic reconstruction showing the distribution of the CAMP. Can you provide a reference?

What are the thin grey dashed lines? What are the thicker grey lines? Can you 'clean up' this figure and eliminate unnecessary information?

What is the base map? I presumed that it shows elevation. What do the colors represent? Would a simple base map work better? The emphasis needs to be on the data not the topography.

Velocity Model Constraints on Crustal Structure

Could you reference the Marzen et al. (2019) paper in this section to help the readers better understand the procedure?

Lines 98 – 101. This sentence does not make sense to me. Also, Reference 30 is about igneous activity on the North Atlantic margin. How does this relate to this sentence?

Lines 101 – 107. One of the key assumptions of the manuscript is that the velocity of the lower crust correlates with the amount of mafic intrusions in the lower crust. Could the authors build a clearer case to support this assumption. What are typical lower crustal velocities in this region? What are reasonable ranges? What are typical velocities of mafic intrusions? As the authors state, supporting evidence includes the distribution of the high-velocity rocks beneath the depocenter of the rift basins and location of the thinnest crust.

Figure 2.

I have some issues with this figure and caption. What are the purple lines beneath the Moho? What does the grey area represent?

Could the authors show the locations of the rift basins on these cross sections? One of the key points is that crustal thickness is least and magmatic thickness is greatest beneath the rift basins. It would be easier to see this relationship if the rift basins were shown.

Note typo in caption ... and (d) and.

Is something wrong with the last sentence of the figure caption? The crustal stretching estimates are only for part d. In part b, the black line shows the crustal thickness with the mafic intrusions included and the red area shows the range of crustal thickness with the mafic intrusions removed.

Why would the black lines go above the red areas in parts b and d for Line 2? The crustal thickness would never have been less with the mafic intrusions included than without the mafic intrusions. For Line 2, the authors show intrusion thicknesses only to the SE of the 250 km marker (part c). Why is there a red area to the NW of the 250 km marker on parts b and d?

South Georgia Rift Magmatism

Lines 115 – 120. Very interesting.

Line 120. Is it lithospheric thinning or crustal thinning or both?

Line 134. ... calculated cumulative intrusion thickness

Also, the use of the informal work 'so' writing is generally discouraged in scientific writing. Can you change the sentence to ... Because the composition and velocity of the magmatic intrusions is unknown, we calculated

Lines 142 – 143. Did you really show a spatial correlation between elevated lower crustal velocities and rift basins? Is this true on Line 2 which also crosses a rift basin? Why would the magnitude of crustal intrusions be so much larger beneath the rift basin on Line 1 than that on Line 2? The authors need to mention the correlation of elevated lower crustal velocities and the rift basins on Line 2. Showing the locations of the rift basins on Figure 2 would help with this discussion.

Lines 149 – 151. Crustal thicknesses were likely greater ... than what? Than the assumed 45 km? By conservative, do you mean smaller than what it would be for a thicker crust?

Lines 155 – 158. There is little control on the exact timing of breakup in the SE US. Can you include more recent plate-kinematic models for the opening of the Atlantic Ocean in addition to Reference 37 from 1986? The newer models favor an older age (closer to 190 Ma), but the age of breakup (and the age of the SDRs) is poorly constrained.

Also, here you are implying that the elevated lower crustal velocities on Line 2 are associated with the proximity to the site of breakup not to the overlying rift basin. This contradicts some of your statements about the correlation of synrift sediment thickness and the thickness of the magmatic intrusions in the lower crust.

Lines 162 – 164. Is there a positive correlation between synrift sediment thickness and the thickness of the magmatic intrusions in the lower crust? Again, it needs to be clearly shown and stated that the lower crust beneath the rift basins on Line 1 AND Line 2 has elevated velocities. Earlier in the paper, the authors imply that the elevated velocities on Line 2 are related to breakup processes.

Line 165 – Is this synrift sediment thickness, or do you mean the thickness of the entire synrift section (including sills and flows)?

Line 167 – Do you mean lower crustal V_p of 7.5 and 7.2 km/s, or do you mean the velocity of the crustal intrusions?

Condition during Magma Generation and Emplacement

Line 177. Note misspelling of Emplacement

Lines 179 – 181. Does crustal thinning necessary equate with mantle thinning? If the stretching on the margin was depth dependent with more crustal thinning onshore and more mantle thinning offshore, would this affect your calculations? The depth-dependent stretching that you describe in the Methods section appears to have more mantle stretching beneath the rift basins. Could the opposite be true?

Lines 188 – 203. With the decompression melting model, would intrusion of the lower crust occur during rifting, not just during CAMP activity? It seems like the authors are relating the intrusions to

rifting in this section, but in previous sections of the model they are relating the intrusions to CAMP. Can you clarify this? [Never mind, I just read the next section after Figure 3. Perhaps, you need to warn the readers that the decompression melting models seem to contradict some of the previous statements.]

What would happen if the source of the crustal intrusions was not directly below the rift basins but further to the southeast near the site of breakup? Could there have been lateral flow of deep magma from the southeast to the northwest beneath the rift basins? We tend to assume that everything is vertical. But, this might not be true.

Some new age dates from New England imply that igneous activity (dikes) preceded CAMP by about 20 million years. I suggest that the authors review some of the recent papers on the Coastal New England magmatic province. These dikes would have been 5 to 10 kilometers below the Earth's surface during rifting.

The South Georgia rift basin complex, like many other ENAM (eastern North American) rift basins, is offset from the site of breakup by hundreds of kilometers. Is it possible that the volume of CAMP-related crustal intrusions was considerably greater in the offshore region which underwent greater extension and was closer to the site of breakup? How do the observations relate to the seaward dipping reflectors offshore? These might be important questions to address in this section.

Figure 3.

Should the vertical axis be ... Cumulative Thickness of Igneous Intrusions?

Methods -- Igneous Intrusion Thickness Calculation from Seismic Velocities

Line 279. Define each term of the equation.

Line 283. ... increases in crustal velocity at these depths reflect (no s)

Lines 288. What do you mean by ... the velocity of intruded crust was estimated at 7.2-7.5 km/s? Do you mean the velocity of the material that intruded the crust? This is very different than the velocity of the intruded crust.

Reviewer #3 (Remarks to the Author):

This manuscript sets out with a bold hypothesis – that the volume of the CAMP (and hence its formation mechanism) can be constrained by two closely spaced wide-angle seismic profiles. The CAMP is an especially interesting LIP as it has a wide surface expression yet lacks the plume tail and high temperature volcanics that are thought characteristics of plume sources.

One of my concerns is that the context of the CAMP was rather superficial. For example, previous seismic experiments (particularly those across the volcanic margins of the eastern US e.g. Carolina Trough, Blake Plateau) in the area were not sufficiently referenced. These studies identified strong lateral (along-strike) variations in lower-crustal magmatism that is relevant to the discussion presented here. There are also other hotspot influences in the region (Bahama) that should have been included. In a similar manner the magmatic volumes calculated were described as “modest” but no other comparisons to other LIP provinces were given.

A second concern is the assertion that the two profiles presented lie "at the (CAMP) event's centre" (line 29). I felt this was poorly justified. Yes, we have a pink outline for the CAMP in the inset of Fig 1, but without any other details such as mapped dyke locations or previous temperature estimates it is hard to conclude any spatial changes in magmatic activity from which a "centre" could be estimated. In any case the concept of a "centre" to such a province seems to assume axisymmetric plume head – what if instead the source is an elongated subducted slab? Because of these issues I had difficulties extrapolating the results of the two wide-angle profiles to the CAMP province as a whole. Possibly this issue could be fixed by a carefully drafted tectonic setting figure and text.

My third concern is about the analysis itself. The sedimentary thickness is an important observation, yet it wasn't possible to assess how well constrained this was. In the supplement there is mention of "other data" such as COCORP reflection seismic - but no details or references are given. Similarly, the lower crustal seismic velocities are critical. As is typical in wide-angle work the critical lower crustal region suffers from a lack of turning rays. The authors are clearly aware of this as they attempt to place uncertainties on their reliance on PmP. However, it would also be good to show a checkerboard. We should also have a fig s4 and s5 for the other seismic line.

Minor points

- Correct the use of capitals for proper nouns throughout e.g., eastern North and South America; South Georgia Basin.
- Inconsistent Line 31 "one of the most"; line 15 "the most"
- Fig2 caption. State what velocity contour range was used to define the mafic intrusion thickness (7.2-7.5 km/s, line 135?). Not convinced part d adds anything. Why does line 1b,d have red ribbons at $x < 250$ km given line 1c doesn't? It is hard to see the sediment thickness variations discussed in the text – and given the correlation with the magmatism is central to the argument presented perhaps these should have been plotted as a separate graph?
- Line 305. This is very vague – give some numbers!
- The reference list is full of errors e.g. 4 and 43 are from a book, but neither reference is correct (and to make matters worse are inconsistent); 10 typo Tectonophysics2; 51 C.A.
- Fig S-2. It is odd given you want the reader to compare gradients not to have identical axis integrals (presumably the distance one is the same in both plots but it is hard to see this). Also it would have been much better to have used offset rather than distance to allow the reader to compare more easily. What is the topography along the profile (Fig S-1 is missing a topo scale).

Reviewer #1 (Remarks to the Author):

This is a very interesting, well-written, important manuscript about geophysical data on the crustal structure in the south-eastern USA (Georgia rift). The authors show quite convincingly that the deep crust was intruded by basic magmas in at least parts of the crustal roots of the South Georgia Rift. The thickness of such basic layers is does generally not exceed 2 km and is sometimes much thinner or even absent. The authors interpret these mafic layers as related to the CAMP event and consider that the generally low volumes of such rocks indicate that CAMP magmatism formed from a not very hot mantle. Apparently, there is also a thinning of the deep crustal gabbros from West to East, i.e. towards the central Atlantic margin. These findings are very important, as they constrain for the first time the deep crustal roots of CAMP magmatism (however, intermediate crust intrusions are present in west Africa, i.e. Sierra Leone; Callegaro et al., J. Petrology, 2017) and bear important

constraints on the genesis of this magmatic province and on its relation with the break-up of Pangea and the opening of the Atlantic ocean. The conclusions are compatible with what we know from the rest of CAMP. Figures are very nice. The adopted methods are excellent, as far as I can understand as a non-specialist (geophysicist).

The only part of the manuscript, which seems little convincing to me is the discussion about the timing of this basic magmatism. The authors suggest that this magmatism may be significantly older (by at least 4 million years) than the rest of CAMP. This is based chiefly on the fact that sedimentation in the South Georgia Rift stopped by the end of the Norian (ca. 205 Ma). However, the evidence for such stop of sedimentation and thus rifting is not clear from the text. The authors should briefly discuss geological evidences of this. The authors seem also to propose that the deeply intruded magmas may be significantly older than those erupted or intruded as shallow sills or dykes. Such idea is not very convincing, as in any case a shallow magma requires the presence of deep intrusions. I.e., CAMP basalts are evolved melts resulting from at least 40-50% fractionation of a primary mantle melt. The fractionated mass and volume intruded in the intermediate or deep crust and may well be the parts of the basic intrusions identified by the authors.

All the best,

Andrea Marzoli

We have added the Callegaro et al., 2017 reference to the introduction when we refer to the limited constraints on CAMP in the mid- to lower crust.

To address the feedback that we better explain the evidence that South Georgia rifting predated magmatism, we have revised the text to more thoroughly explain the available geologic constraints:

“The correlation between magmatic intrusions and the Triassic South Georgia Basin evident in velocity models is surprising because multiple geological constraints suggest that magmatism was emplaced after, not during, the formation of the South Georgia Rift. First, CAMP-dated sills and dikes crosscut synrift strata and orogenic structures in the shallow subsurface^{33,55}. Second, flows in the Southeastern US are only found in the postrift section^{35,36}. Third, the orientation of

CAMP-dated dikes (Fig. 1) is incompatible with the NW-SE minimum horizontal stress orientation consistent with the orientation of basin structures³⁴.”

One of the biggest surprises and primary results of our study is that the distribution of interpreted mafic intrusions at depth appears to correlate with the rift basins, which are older than CAMP as described above. This is a surprise since the shallow manifestation of CAMP does not correlate with rift basins in this region. We have explored many possible models to reconcile this correlation with the different ages of the SGRB and CAMP. Of these, we think the most likely scenario that is most consistent with existing observations, including those presented here, is that the conditions that promoted CAMP (insulation beneath a super continent and/or mantle enrichment) enabled magmatic intrusions beneath the South Georgia Rift during its formation, and that a later stress change enabled these magmas to travel to the surface during a short window. While this explanation accounts for the correlation of magmatism and crustal thinning, it requires that magmatism at depth formed earlier and over a longer time period than shallow CAMP sills and lavas, which we emplaced over a short time period. The alternative explanation is that there was some synrift magmatism beneath the SGRB, but CAMP was caused by a different “event” later (like delamination) and that CAMP intrusions at depth may have been focused beneath the basin by lithospheric topography created during rifting. We cannot distinguish between these possibilities based on our data. However, in either explanation, our main results hold true: 1) CAMP magmatism was limited at depth and 2) the distribution of magmatism in the crust (be it from CAMP or earlier synrift magmatism) appears to be localized by lithospheric thinning associated with the SGRB. The beginning of the Discussion has been revised to the text below:

“The combined geophysical, geological, and geochemical observations in the Southeastern US can be explained by magmatic intrusions sourced by decompression melting during continental extension between 230-205 Ma but then later reaching the surface at ~201 Ma. The geochemistry of CAMP magmas indicates 30-50% fractionation from a mantle-derived magma¹⁸, which is consistent with our new constraints on the amount of magmatism at depth. The biggest challenge with reconciling available constraints is the timing of emplacement of magmatism. The amount of magmatism emplaced in the lower crust has a similar volume and spatial distribution to what would be expected for decompression melting during rifting (~230-205 Ma) of a warm mantle, but rifting in the Southeastern US took place earlier and over a longer time period than the short ~0.6 m.y. duration of CAMP intrusions at 201 Ma. Explaining this timing difference requires that some mechanism, such as a change in stress state⁶⁰ (i.e., associated with extension at the modern rifted margin), allowed magmatism generated by synrift decompression melting below the South Georgia rift to be emplaced at the Earth’s surface over a short period of time. In some active rift systems, evidence for magmatism is observed at depth even though volcanism is not observed at the surface⁶¹. In the South Georgia Rift, the relatively thick lithosphere after limited continental extension may have delayed transport of magmatism to the Earth’s surface.”

Reviewer #2 (Remarks to the Author):

I have reviewed the manuscript by Marzen and colleagues entitled, “Limited and Localized Magmatism at the Heart of the Central Atlantic Magmatic Province.”

The authors have used velocity models on two seismic transects from the southeastern United States to constrain the volume and distribution of CAMP-related crustal intrusions. They conclude that the volume of igneous intrusions is small and that the intrusive activity was focused beneath depocenters of the South Georgia rift basin complex. Thus, crustal thinning associated with Triassic rifting affected the magnitude and distribution of CAMP-related intrusive activity.

It is very interesting to see convincing (and quantifiable) evidence of CAMP-related crustal intrusions beneath an onshore rift basin on the margin of eastern North America. It is also very interesting to see that the shallow distribution of CAMP-related igneous activity is much broader than that in the lower crust.

As with most research, this work answers some key questions and raises many more. I think that the research will be of interest to geologists and geophysicists who study rifted margins worldwide, not just the rifted margin of eastern North America.

As discussed below, I found some parts of the manuscript difficult to read and/or understand. Also, some of the figures need improvement and clarity.

Figure 1.

I have some issues with this figure and caption. The authors need to describe the insert on the bottom right. I presume that this is a Triassic reconstruction showing the distribution of the CAMP. Can you provide a reference?

The inset does indeed show a Triassic reconstruction and the distribution of CAMP. The figure caption has been edited to include the following text and reference to Marzoli et al., 2018. *“Inset shows the approximate extent of CAMP in red over a reconstruction of Pangea¹⁸.”*

What are the thin grey dashed lines? What are the thicker grey lines? Can you ‘clean up’ this figure and eliminate unnecessary information?

The thin grey dashed lines and the thicker grey lines within the basin show transfer zones and faults interpreted in the basin model (Heffner, 2013). The legend of the map has been revised to clarify what these lines represent and the color of the diabase dikes has been changed to brown to distinguish them from the transfer zones and faults.

What is the base map? I presumed that it shows elevation. What do the colors represent? Would a simple base map work better? The emphasis needs to be on the data not the topography.

We have added a color scale for the elevation data to the legend to explain the basemap. Including these data in the one map of the main manuscript is helpful to readers because the transition from the Appalachians to the Coastal Plain is strongly related to the location of the rift basins that are a focus of this manuscript.

Velocity Model Constraints on Crustal Structure

Could you reference the Marzen et al. (2019) paper in this section to help the readers better understand the procedure?

The Marzen et al. 2019 reference has been added to the first paragraph of this section.

Lines 98 – 101. This sentence does not make sense to me. Also, Reference 30 is about igneous activity on the North Atlantic margin. How does this relate to this sentence?

The text has been revised to the following to clarify this point:

“The observed variations are within a single crustal terrane³⁸, so contrasts between crustal terranes cannot explain our observations (Fig. 1). We thus interpret these localized increases in lower crustal velocity as the addition of mafic magmatic intrusions^{46,47}.”

The references in question refer to literature about seismic velocities of non-mafic versus mafic lower crust (Hacker et al., 2015) and how elevated lower crustal seismic velocities have been similarly interpreted as the addition of mafic magmatic intrusions (White et al., 2008).

Lines 101 – 107. One of the key assumptions of the manuscript is that the velocity of the lower crust correlates with the amount of mafic intrusions in the lower crust. Could the authors build a clearer case to support this assumption. What are typical lower crustal velocities in this region? What are reasonable ranges? What are typical velocities of mafic intrusions? As the authors state, supporting evidence includes the distribution of the high-velocity rocks beneath the depocenter of the rift basins and location of the thinnest crust.

We have revised this paragraph in the text to clarify the relationship between elevated velocities and mafic magmatic intrusions, typical lower crustal velocities in the region, and typical mafic magmatic intrusion velocities:

“Both seismic profiles indicate limited and localized regions of elevated >7.0 km/s lower crustal velocities in the South Georgia Rift (Fig. 2a). The most likely explanation for changes in lower crustal velocity in this region is changes in composition. The observed variations are within a single crustal terrane³⁸, so contrasts between crustal terranes cannot explain our observations (Fig. 1). We thus interpret these localized increases in lower crustal velocity as the addition of mafic magmatic intrusions^{46,47}. Seismic refraction measurements from offshore of eastern North America indicate that mafic lower crust velocities typically range from 7.2-7.5 km/s^{24,48–50}, which is similar to the highest lower crustal velocities directly constrained by rays that turn in the lower crust in the SUGAR velocity models. These velocities also encompass different intrusion compositions predicted for different depths of melting⁵¹. In contrast to velocities of

mafic intrusions, unmodified continental lower crust is typically ~6.8 km/s⁵², and well-constrained lower crustal velocities on SUGAR Line 2 indicate that the lower crustal velocities northwest of the South Georgia Rift Basin are ~6.75 km/s⁴⁴.”

Figure 2.

I have some issues with this figure and caption. What are the purple lines beneath the Moho? What does the grey area represent?

The purple lines beneath the Moho show where Pn rays sample the upper mantle, and their color reflects their mantle velocities ~8 km/s (color bar). This is also highlighted by the purple “Moho” label placed at Moho depths in the space between the two velocity models.

The following edit has been made to the caption to clarify the fact that the grey color is the mask we place on our velocity model where there is no ray coverage and thus velocities are not constrained:

“(a) V_P models with 0.5 km/s contour interval for Line 1 (left) and Line 2 (right). Black triangles indicate shot locations. Areas with no ray coverage are masked in grey.”

Could the authors show the locations of the rift basins on these cross sections? One of the key points is that crustal thickness is least and magmatic thickness is greatest beneath the rift basins. It would be easier to see this relationship if the rift basins were shown.

We have revised the figure to include a new panel that shows the thickness of the rift basins based on the basin model of Heffner (2013) (also show in Fig. 1).

Note typo in caption ... and (d) and.

Fixed typo

Is something wrong with the last sentence of the figure caption? The crustal stretching estimates are only for part d. In part b, the black line shows the crustal thickness with the mafic intrusions included and the red area shows the range of crustal thickness with the mafic intrusions removed.

The figure caption has been revised to no longer refer to sub-plot b (now c) in reference to crustal stretching.

Why would the black lines go above the red areas in parts b and d for Line 2? The crustal thickness would never have been less with the mafic intrusions included than without the mafic intrusions. For Line 2, the authors show intrusion thicknesses only to the SE of the 250 km marker (part c). Why is there a red area to the NW of the 250 km marker on parts b and d?

In areas where the average lower crustal velocity is less than the reference velocity of 6.75 km/s, there were some slightly negative values for intruded magmatism as a by-product of the

calculation. This caused intrusion thicknesses NW of 250 km to be negative and the crustal thickness to be slightly less with the mafic intrusions included. We agree with the reviewer that this is not realistic and have adjusted the calculation so there are never negative intrusion thicknesses by setting those values to zero. Where the intrusion thicknesses are 0, the crustal thickness is the same with and without intrusions.

To make the calculation clear to readers, the equation and text in the “Igneous Intrusion Thickness From Seismic Velocities” section have been revised to reflect this change.

We have also updated Figure 3 and Supplementary figures S-6, S-7, S-8, Supplementary Table S-4, and the manuscript estimates of total intrusion thickness based on this minor fix to our estimation of intrusion thickness. The changes in values were small relative to the level of precision (i.e. 125,000 versus 127,000 km³) and did not change the interpretation or implications of the results.

South Georgia Rift Magmatism

Lines 115 – 120. Very interesting.

Line 120. Is it lithospheric thinning or crustal thinning or both?

We expect that the whole lithosphere, including the crust and mantle lithosphere, thinned during the formation of the South Georgia Rift. We assume a simple model of uniform thinning of the crust and mantle lithosphere, in the absence of any direct constraints on variations in thinning with depth.

Line 134. ... calculated cumulative intrusion thickness

Also, the use of the informal work ‘so’ writing is generally discouraged in scientific writing. Can you change the sentence to ... Because the composition and velocity of the magmatic intrusions is unknown, we calculated

Sentence changed to the revision suggested.

Lines 142 – 143. Did you really show a spatial correction between elevated lower crustal velocities and rift basins? Is this true on Line 2 which also crosses a rift basin? Why would the magnitude of crustal intrusions be so much larger beneath the rift basin on Line 1 than that on Line 2? The authors need to mention the correlation of elevated lower crustal velocities and the rift basins on Line 2. Showing the locations of the rift basins on Figure 2 would help with this discussion.

Panels showing the thickness of the synrift sedimentary fill across both seismic lines have been added to Figure 2. We estimate thicker intrusions beneath Line 1 where there appears to be more concentrated crustal stretching and a thicker synrift sedimentary fill than on Line 2. The following sentence has been revised in the text to better explain this contrast:

“Rift basin sedimentary fill is thicker on Line 1 than on Line 2, which is consistent with seismic reflection imaging and core data from the west versus east South Georgia Rift (Fig. 1)^{24,52}. “

Based on the synrift thicknesses shown in Figure 2 and analysis described in the methods section “South Georgia Rift Magma Volume Calculation”, the limited magmatic intrusions beneath Line 2 are consistent with the limited synrift sedimentary fill thicknesses in the Riddleville sub-basin (0 expected magmatism for < 1 km synrift thickness). The synrift sedimentary fill in the Kibbee sub-basin is thicker and lower crustal velocities are correspondingly higher.

Lines 149 – 151. Crustal thicknesses were likely greater ... than what? Than the assumed 45 km? By conservative, do you mean smaller than what it would be for a thicker crust?

Yes, and this text has been revised to clarify these points.

Lines 155 – 158. There is little control on the exact timing of breakup in the SE US. Can you include more recent plate-kinematic models for the opening of the Atlantic Ocean in addition to Reference 37 from 1986? The newer models favor an older age (closer to 190 Ma), but the age of breakup (and the age of the SDRs) is poorly constrained.

Also, here you are implying that the elevated lower crustal velocities on Line 2 are associated with the proximity to the site of breakup not to the overlying rift basin. This contradicts some of your statements about the correlation of synrift sediment thickness and the thickness of the magmatic intrusions in the lower crust.

Based on the geometry of crustal thinning and inferred lower crustal intrusion volumes, the southern portion of Line 2 was likely impacted by two major rifting events – rifting to form the South Georgia Basin and breakup of Pangea. The text has been revised to clarify this point. The text below and in the introduction has also been revised to better explain (1) the uncertainties in timing of offshore breakup, (2) the range in dates for continental break-up:

“The patterns of crustal thinning, magmatic addition, and extent of the South Georgia Rift Basin on Line 2 suggest that two episodes of rifting concentrated crustal thinning along different portions of the transect. We infer that thinning of the crust beneath the South Georgia Basin (Fig. 1) was associated with the extension to form this rift basin and that the southeastern portion of Line 2 was likely additionally thinned later during the breakup of Pangea ~175-195 Ma²⁶⁻²⁸. Although the exact timing of continental breakup is uncertain, we would expect this successful rifting event to concentrate crustal stretching and magmatic intrusions towards the margin. This is consistent with the pattern of crustal stretching and magmatic addition from 280 km distance to the southeastern end of the transect, which increases towards the rifted margin and is not centered beneath the South Georgia Rift. To focus our analysis on the South Georgia Rift, the analysis below excludes the southeastern portion of Line 2.”

Lines 162 – 164. Is there a positive correlation between synrift sediment thickness and the thickness of the magmatic intrusions in the lower crust? Again, it needs to be clearly shown and stated that the lower crust beneath the rift basins on Line 1 AND Line 2 has elevated velocities. Earlier in the paper, the authors imply that the elevated velocities on Line 2 are related to

breakup processes.

We believe that the revised text in response to the reviewer's previous comment will clarify this question. There is a positive correlation between synrift sediment thickness and the thickness of magmatic intrusions of the lower crust on both seismic lines, which is shown in supplementary figure S-8. On the southeastern end of Line 2, which approaches the rifted margin and site of continental break-up, we propose that the crust was most likely modified largely by the later breakup of Pangea and not South Georgia rifting. This part of Line 2 is south of the main basins of the South Georgia Rift. The methods section that is referenced in this text includes the following description of how we set up our analysis to be consistent with this interpretation:

"All parts of both seismic lines were used to constrain the calibration except where we do not have resolution of the lower crust and Moho or where the magmatism may be sourced from the breakup of Pangea instead of CAMP at the southeast end of Line 2 (i.e., constraints from Line 1: 50-250 km distance; Line 2: 50-280 km distance)."

Line 165 – Is this synrift sediment thickness, or do you mean the thickness of the entire synrift section (including sills and flows)?

It is challenging to separate synrift sediment fill from flows with the resolution of refraction seismic data. However, available well data have relatively low sill and flow thicknesses compared to sediment fill (e.g. Fig. 2 in Heffner et al., 2012). Mafic layers are typically 10s of m thick while the synrift fill varies from 100s of m to ~4 km thick. Thus, our synrift sediment thickness does include some CAMP sills.

Additionally, the text has been revised to clarify the fact that the mafic magmatic intrusion thicknesses we are referring to are in the lower crust, based on elevated lower crustal seismic velocities.

Line 167 – Do you mean lower crustal V_p of 7.5 and 7.2 km/s, or do you mean the velocity of the crustal intrusions?

This is referring to the velocity of the intrusions, and the text has been revised to clarify that these are assumed intrusion velocities used to estimate the total thickness of lower crustal intrusions.

Condition during Magma Generation and Emplacement

Line 177. Note misspelling of Emplacement

Fixed

Lines 179 – 181. Does crustal thinning necessary equate with mantle thinning? If the stretching on the margin was depth dependent with more crustal thinning onshore and more mantle thinning offshore, would this affect your calculations? The depth-dependent stretching that you describe in the Methods section appears to have more mantle stretching beneath the rift basins. Could the

opposite be true?

Since it is not possible to reconstruct the geometry of the lithosphere-asthenosphere boundary at 200 Ma, we assume simple depth-uniform extension in the main text. More complicated variations in stretching with depth are possible. We considered the case of greater lithospheric thinning compared to crustal stretching inspired by results from the active, magmatic East Africa Rift System, where a combination of lithospheric thinning and thermochemical erosion leads to enhanced lithospheric thinning (e.g., Bastow et al, 2010). In this case, we would predict more magmatism for the same temperature conditions and mantle compositions. We do not think that a lateral offset between crustal thinning and mantle thinning is consistent with our observed localization of crustal thinning and magmatic intrusions beneath the basins, which implies that lithospheric thinning was also concentrated in this region.

Lines 188 – 203. With the decompression melting model, would intrusion of the lower crust occur during rifting, not just during CAMP activity? It seems like the authors are relating the intrusions to rifting in this section, but in previous sections of the model they are relating the intrusions to CAMP. Can you clarify this? [Never mind, I just read the next section after Figure 3. Perhaps, you need to warn the readers that the decompression melting models seem to contradict some of the previous statements.]

We have added the following text to preview the results of the decompression melt modeling section and the rationale for undertaking these rift-related melt models:

“These models provide a means to compare the quantity of lower crustal magmatic intrusions to volumes predicted to be generated during rift-related decompression melting because we cannot constrain the timing or source of lower crustal magmatic intrusions directly.”

We have also revised other text earlier in the paper to anticipate our results and interpretation.

What would happen if the source of the crustal intrusions was not directly below the rift basins but further to the southeast near the site of breakup? Could there have been lateral flow of deep magma from the southeast to the northwest beneath the rift basins? We tend to assume that everything is vertical. But, this might not be true.

While we cannot rule out an extreme case where all the magmatism was generated offshore and flowed west, we would expect magmatism to pool where lithospheric stretching was greatest. In the hypothetical scenario described above, that would be east of the South Georgia Rift (where break-up ultimately occurred). This offset of magmatic intrusions from the rift basin is not observed in our data, particularly on Line 1 where the largest amount of crustal thinning, thickest synrift sedimentary fill, and the most magmatic intrusions are observed/inferred.

Some new age dates from New England imply that igneous activity(dikes) preceded CAMP by about 20 million years. I suggest that the authors review some of the recent papers on the Coastal New England magmatic province. These dikes would have been 5 to 10 kilometers below the Earth's surface during rifting.

We agree with the reviewer that multiple magmatic events have impacted Eastern North America, and available age constraints indicate that the distant Coastal New England magmatic province was synchronous with South Georgia rifting. The absence of any data indicating that similar magmatism occurred in the Southeastern US (e.g. Mazza et al., 2017) does not mean that there was none, but the currently available geological and geochronological constraints lead us to infer that there was no significant syn-rift extrusive magmatism in the Southeastern US: (1) the absence of shallow syn-rift sills or lavas in the region (Hames et al., 2000; Mazza et al., 2017), (2) dike intrusion orientations inconsistent with syn-rift emplacement (Schlische, 2003), and (3) stratigraphic relationships between rift strata and shallow intrusions indicating magmatism was emplaced after rifting (Withjack et al., 2012; McBride, 1989).

We have added this justification for our interpretation to the manuscript text:

“In contrast, all available evidence from the stratigraphic record^{33,55}, dike orientations as an indicator of the stress field³⁴, and intrusion dates in the Southeastern US^{32,56} indicate that near-surface magmatism in the Southeastern US occurred during CAMP ~201 Ma, after South Georgia rifting.”

The South Georgia rift basin complex, like many other ENAM (eastern North American) rift basins, is offset from the site of breakup by hundreds of kilometers. Is it possible that the volume of CAMP-related crustal intrusions was considerably greater in the offshore region which underwent greater extension and was closer to the site of breakup? How do the observations relate to the seaward dipping reflectors offshore? These might be important questions to address in the this section.

We agree with the reviewer that there are a lot of outstanding questions about the relationship between CAMP, ENAM and the South Georgia Rift and we can only speculate about how much CAMP magmatism exists offshore based on other studies. Offshore drilling would help answer these questions, but the most recent interpretations indicate that a lot of the magmatism offshore was emplaced during a prolonged period that continued after CAMP (Davis et al., 2018). We have added text in the manuscript’s introduction to better explain what is known about the timing and location of ENAM relative to the South Georgia Rift and what is known about the timing of offshore magmatism including SDRs (copied below).

“Although extensive magmatism has been imaged on these margins including in the Blake Plateau Basin and the Carolina Trough^{23,24}, the timing and duration of the emplacement of this magmatism is unknown²⁵. Ages between 172 and 200 Ma²⁶⁻³⁰ and emplacement durations up to 6-31 Myr³¹ have been suggested, so it is unclear if offshore magmatism was related to CAMP.”

Figure 3.

Should the vertical axis be ... Cumulative Thickness of Igneous Intrusions?

We agree that this label for the y-axis would be more clear, and have changed it to “Cumulative Igneous Intrusion Thickness (km)”

Methods -- Igneous Intrusion Thickness Calculation from Seismic Velocities

Line 279. Define each term of the equation.

These equation terms are now defined with the following sentence added to the text:
“where Z_{int} is the thickness of mafic intrusions, Z_{tot} is the thickness of the crust below 20 km depth, V_{P-orig} is the reference velocity for the lower crust without intrusions, V_{P-obs} is the observed average lower crustal velocity (below 20 km depth), and V_{P-int} is the assumed velocity of mafic magmatic intrusions.”

Line 283. ... increases in crustal velocity at these depths reflect (no s)

Fixed

Lines 288. What do you mean by ... the velocity of intruded crust was estimated at 7.2-7.5 km/s? Do you mean the velocity of the material that intruded the crust? This is very different than the velocity of the intruded crust.

The text has been revised to clarify that these values refer to the assumed velocity of the material that intruded the lower crust, which is compared with the observed velocities to estimate total magmatic addition.

Reviewer #3 (Remarks to the Author):

This manuscript sets out with a bold hypothesis – that the volume of the CAMP (and hence its formation mechanism) can be constrained by two closely spaced wide-angle seismic profiles. The CAMP is an especially interesting LIP as it has a wide surface expression yet lacks the plume tail and high temperature volcanics that are thought characteristics of plume sources.

One of my concerns is that the context of the CAMP was rather superficial. For example, previous seismic experiments (particularly those across the volcanic margins of the eastern US e.g. Carolina Trough, Blake Plateau) in the area were not sufficiently referenced. These studies identified strong lateral (along-strike) variations in lower-crustal magmatism that is relevant to the discussion presented here. There are also other hotspot influences in the region (Bahama) that should have been included. In a similar manner the magmatic volumes calculated were described as “modest” but no other comparisons to other LIP provinces were given.

This comment has many components, which are addressed individually below:

1. We have added more introductory context and references for CAMP, including introducing different possible relationships between onshore and offshore (e.g. Carolina Trough, Blake Plateau) magmatism. The reason that we did not discuss offshore magmatism in our paper is that its age is poorly constrained, and many authors propose that it was emplaced after CAMP and/or over a longer time period than CAMP. However, we agree that the offshore magmatism needs to be addressed more directly in this manuscript.

“Another uncertainty in estimating the total volume of CAMP is the age and origin of magmatism offshore along the rifted margins of Pangea. Although extensive magmatism has been imaged on these margins including in the Blake Plateau Basin and the Carolina Trough^{23,24}, the timing and duration of the emplacement of this magmatism is unknown²⁵. Ages between 172 and 200 Ma^{26–30} and emplacement durations up to 6–31 Myr³¹ have been suggested, so it is unclear if offshore magmatism was related to CAMP.”

We agree that the observations of along-strike changes in the volume of magmatism in recent studies (e.g., Shuck et al, 2019) are very interesting and potentially relevant to our result . However, because of the limited constraints on the timing of offshore magmatism and the relationship between onshore and offshore magmatism, summarized above, we do not think that additional discussion of onshore/offshore similarities would enhance the text.

2. To provide context for the volume of CAMP, including our new constraints, we have added further comparison of the volume of CAMP to that of other LIPs in terms of the average thickness of magmatic intrusions associated with CAMP and other LIPs, that can be compared directly with our observations. The statement below has been added.

“CAMP is thought to be a relatively low-volume LIP because the average thickness of magmatic addition of ~0.3 km¹⁸ is far less than estimates for other major LIPs, which are typically ~1 km or greater^{19,20}.”

We have also added the following statement to the discussion on the volume of magmatism in order to provide context for the results.

“This implies an average melt thickness of ~1.5 km in the South Georgia Rift Basin, where magmatism appears to be particularly concentrated compared to areas within the aerial extent of CAMP but outside of the rift basin.”

3. The reviewer indicates that hotspot influences need to be discussed more, particularly the Bahama hotspot. The introduction of our paper references multiple studies that indicate that CAMP itself is not plume-related (*“Although early work hypothesized CAMP might have been caused by a mantle plume⁷⁻⁹, the absence of a plume trail¹⁰, relatively cool mantle temperatures estimated for CAMP¹¹, and isotopic and trace element characteristics¹² argue against a plume source and distinguish CAMP from other large igneous provinces.”*). If a later plume (e.g. Bahama) caused further magmatic addition to the crust in our study area, it would imply an even smaller magmatic contribution from CAMP, further strengthening our hypothesis that CAMP was a low-volume magmatic event.

Although a hotspot (Fernando de Noronha) associated with the Southeastern US and the Bahamas had been previously hypothesized by some authors to have contributed to the breakup of Pangea (Morgan, 1983, *Developments in Geotectonics*; Courtillot et al., 1999, *Earth and Planetary Science Letters*), there is limited evidence suggesting that the hot spot is deeply rooted (Courtillot et al., 2003, *Earth and Planetary Science Letters*) or that it even exists, given magmatism ages in Brazil over the suspected plume’s track (Knesel et al., 2011, *Earth and Planetary Science Letters*) and lack of observable anomaly in global seismic tomography (Montelli et al., 2006, *Geochemistry, Geophysics, Geosystems*). For these reasons, we have not included further discussion of the Bahamas and plume influences in the manuscript.

“centre” (line 29). I felt this was poorly justified. Yes, we have a pink outline for the CAMP in the inset of Fig 1, but without any other details such as mapped dyke locations or previous temperature estimates it is hard to conclude any spatial changes in magmatic activity from which a “centre” could be estimated. In any case the concept of a “centre” to such a province seems to assume axisymmetric plume head – what if instead the source is an elongated subducted slab? Because of these issues I had difficulties extrapolating the results of the two wide-angle profiles to the CAMP province as a whole. Possibly this issue could be fixed by a carefully drafted tectonic setting figure and text.

We have changed the manuscript title and revised the abstract to remove the abovementioned phrase, and agree with this reviewer that the concept of the center of a magmatic event has different implications depending on the source of the magmatism (plume or elongated slab). We have limited references to centrality throughout the text, and our main intention is to be clear to readers that these seismic transects are well within the known extent of CAMP magmatism (shown in Fig. 1).

My third concern is about the analysis itself. The sedimentary thickness is an important observation, yet it wasn't possible to assess how well constrained this was. In the supplement there is mention of "other data" such as COCORP reflection seismic - but no details or references are given. Similarly, the lower crustal seismic velocities are critical. As is typical in wide-angle work the critical lower crustal region suffers from a lack of turning rays. The authors are clearly aware of this as they attempt to place uncertainties on their reliance on PmP. However, it would also be good to show a checkerboard. We should also have a fig s4 and s5 for the other seismic line.

We have added the following text to the supplementary material to address the reviewer's interest in more information on constraints on sedimentary thickness, but emphasize that a complete description of the development of the isopach model is presented in Heffner (2013).

"A detailed description of the constraints and methods behind the South Georgia Basin model used in this manuscript is published in Chapter 3 of Heffner (2013) and briefly summarized here. This is the first isopach of sedimentary fill in the South Georgia Basin developed by integrating seismic reflection data (Domoracki, 1995; Petersen et al., 1984; Cook et al., 1981; Nelson et al., 1985; Behrendt, 1985; McBride, 1991; Akintunde et al., 2013; Hamilton et al., 1983; Schilt et al., 1983; Yantis et al., 1983; Chapman and Beale, 2010; Behrendt et al., 1983), seismic refraction data (Bonini and Woollard, 1960; Ackermann, 1983; Amick, 1979; Pooley, 1960; Smith, 1982; Woollard, 1957, Cook et al., 1981), and well data (Falls and Prowell, 2001; Applin, 1951; SCDNR; Cumbest et al., 1992; Aadland et al., 1995; Barnett et al., 1975; Snipes et al., 1995; Costain et al., 1986; Marine and Siple, 1974; Steele and Colquhoun, 1985; Ball et al., 1988; Gohn et al., 1983; Gohn, 1983; Scholle, 1979; Chowns and Williams, 1983; Falls, 1994; Herrick, 1961; Applin and Applin, 1964; McFadden et al., 1986; Milton and Hurst, 1965; Neathery and Thomas, 1975; Gellici, 2007; Mansfield, 1937; Dillon and Popenoe, 1988; additional log data and core observations) throughout the Coastal Plain of the southeastern U.S. Regional 2-D deep seismic reflection profiles from COCORP and SEISDATA were of greatest value in constraining sedimentary basin fill thickness based on subsurface stratal geometries. In some cases, interpretations of these geometries were compromised by the presence of high-impedance mafic sills within the basin stratigraphy. Legacy petroleum exploration wells were instrumental in (1) substantiating the presence of continental redbeds of inferred Triassic age beneath the Coastal Plain unconformity, and (2) in limited cases, constraining the base of these deposits for thickness estimates. The thickness of the deeper, relatively high velocity (~4-5.5 km/s) sedimentary layer from the SUGAR velocity models corresponds to first order with the thickness of Heffner's South Georgia Basin isopach model. The two models diverge, though, to the southeast. The basin isopach model shows the South Georgia Basin thinning to the SE, but the fast layer of SUGAR sediments remains thick to the southeastern end of both seismic lines. These fast sediments to the SE on both seismic lines were likely deposited after South Georgia rifting, but may be older and more compacted than coastal plain sediments, resulting in higher seismic velocities."

We added versions of figure S-4 and S-5 for Line 2 (note that figures have been re-numbered in resubmission) so there are now Supplementary Figures from a tradeoff test for lower crustal velocity/Moho depth for Line 2 in the supplementary material. The tests from Lines 1 and 2

show that the Moho can vary by ~0.75 km with a corresponding velocity change of ~0.05 km/s and still fit the data reasonably well. We attribute this relatively tight constraint on model-wide lower crustal velocities to the combination of lower crustal refractions (Pg), Moho reflections (PmP), and upper mantle refractions (Pn). As a result, the differences between the two profiles, which are the primary observation underlying the manuscript, are significant.

As the reviewer points out, the lower crust is a challenging portion of the model to image with high resolution. While the tradeoff tests described above demonstrate that we can constrain the bulk lower crustal velocity well, we cannot resolve variations at scales of 10s of km. We have added the text below to our Methods section. Our manuscript focuses on the implications of the well-resolved differences between SUGAR Line 1 (high lower crustal velocities, thicker syn-rift sedimentary fill) and SUGAR Line 2 (relatively low lower crustal velocities, thinner syn-rift sedimentary fill).

“The deeper portions of the velocity models including the lower crust are the most challenging to resolve, and there are tradeoffs between increasing crustal thickness and increasing lower crustal velocity. In order to evaluate our velocity model resolution in the lower crust, we evaluated model misfit associated with perturbations in lower crustal velocity and Moho depth (Supplementary Note 1). These tradeoff tests show that the velocity of the lower crust can only be perturbed by up to ~0.05 km/s without increasing the χ^2 fit to the data beyond an acceptable level. The data, however, do not resolve the precise dimensions and locations of localized lower crustal velocity perturbations on the scale of 10’s of km. In addition, our velocity models are most sensitive to perturbations in lower crustal velocity and Moho depth in the central portions of each seismic line where reversed ray coverage is most abundant. In summary, these velocity models are sensitive to overall lower crustal velocity but cannot resolve smaller scale variations in lower crustal velocity. The basis of our result is the large-scale differences in lower crustal velocity and crustal thickness between SUGAR Lines 1 and 2, which are well-resolved.”

Minor points

- Correct the use of capitals for proper nouns throughout e.g., eastern North and South America; South Georgia Basin.

We have revised the capitalization in the manuscript text.

- Inconsistent Line 31 “one of the most”; line 15 “the most”

Text has been revised for consistency:

“The Central Atlantic Magmatic Province (CAMP) is the most aerially extensive but one of the most poorly understood Large Igneous Provinces (LIP) in Earth’s history.”

- Fig2 caption. State what velocity contour range was used to define the mafic intrusion thickness (7.2-7.5 km/s, line 135?).

We believe this revision would be misleading because the thickness of mafic intrusions is not the thickness of the crust in the 7.2 – 7.5 km/s range. The way in which this is calculated is based on

a linear mixing equation (e.g. White et al., 2008) and the relevant equation and methods are described in detail in the “Igneous Intrusion Thickness Calculations From Seismic Velocities”. We have added a reference to the Methods section in the caption to clarify this point.

Not convinced part d adds anything.

We include the crustal stretching factor panel in B because crustal stretching factor plays an important role in Figure 3 (as the X axis) and crustal stretching values provide an indication of the extent to which the rift has developed in comparison to other rifts.

Why does line 1b,d have red ribbons at $x < 250$ km given line 1c doesn't?

Reviewer #2 had a similar concern, so the explanation from above is copied here:

In areas where the average lower crustal velocity is less than the reference velocity of 6.75 km/s, there were some slightly negative values for intruded magmatism as a by-product of the calculation. This caused intrusion thicknesses NW of 250 km to fall below the X axis and the crustal thickness to be slightly less with the mafic intrusions included. We agree with the reviewer that this is not realistic and have since adjusted the calculation so there are never negative intrusion thicknesses by setting those values to zero. Where the intrusion thicknesses are 0, the crustal thickness is the same with and without intrusions.

To make the calculation clear to readers, the equation and text in the “Igneous Intrusion Thickness From Seismic Velocities” section have been revised to reflect this change.

It is hard to see the sediment thickness variations discussed in the text – and given the correlation with the magmatism is central to the argument presented perhaps these should have been plotted as a separate graph?

We agree. The sediment thicknesses from the basin isopach model of Heffner (2013) have been added as a new subplot in Figure 2.

- Line 305. This is very vague – give some numbers!

We agree. Because the volatile content in CAMP magmas is not well known, we have removed this statement from the text.

- The reference list is full of errors e.g. 4 and 43 are from a book, but neither reference is correct (and to make matters worse are inconsistent); 10 typo Tectonophysics2; 51 C.A.

We have revised the references pointed out here, and have also checked and added more complete information to other book references.

- Fig S-2. It is odd given you want the reader to compare gradients not to have identical axis integrals (presumably the distance one is the same in both plots but it is hard to see this). Also it

would have been much better to have used offset rather than distance to allow the reader to compare more easily. What is the topography along the profile (Fig S-1 is missing a topo scale).

We have modified this plot to show both sections at the same scale and to show source-receiver offset rather than distance. A topo scale has been added to Fig. S1, and topography profiles have been added to Fig. S2.

REVIEWERS' COMMENTS:

Reviewer #1 (Remarks to the Author):

The authors answered to the main comments of the reviewers, as much as possible. The manuscript is significantly improved. I don't completely agree with some points of the author's interpretation (i.e., I still think the intrusions should be close to 201 Ma in age), however I think the results of this work are important and I am definitely looking forward to see them published. This paper is certainly contributing a lot to our understanding of the CAMP and of the break-up of Pangea.

All the best,

Andrea Marzoli

Reviewer #2 (Remarks to the Author):

The authors have done an excellent job in honoring my comments and suggestions that I made in my review. Nice paper!

Martha Withjack

BTW, I found two typos.

Line 23 -- Change to ... lower crustal magma

Line 68 -- Delete ... South Georgia rifting or is something missing here

REVIEWERS' COMMENTS:

Reviewer #1 (Remarks to the Author):

The authors answered to the main comments of the reviewers, as much as possible. The manuscript is significantly improved. I don't completely agree with some points of the author's interpretation (i.e., I still think the intrusions should be close to 201 Ma in age), however I think the results of this work are important and I am definitely looking forward to see them published. This paper is certainly contributing a lot to our understanding of the CAMP and of the break-up of Pangea.

All the best,

Andrea Marzoli

Reviewer #2 (Remarks to the Author):

The authors have done an excellent job in honoring my comments and suggestions that I made in my review. Nice paper!

Martha Withjack

BTW, I found two typos.

Line 23 -- Change to ... lower crustal magma

Fixed – to “Lower crustal magmatism is concentrated”

Line 68 -- Delete ... South Georgia rifting or is something missing here

Deleted South Georgia rifting